# Seagrass community-level controls over organic carbon storage are constrained by geophysical attributes within meadows of Zanzibar, Tanzania.

E. Fay Belshe[1,2], Dieuwke Hoeijmakers[1,2], Natalia Herran[1,2], Matern Mtolera[3], Mirta Teichberg[1]

[1]Leibniz Centre for Tropical Marine Research, Fahrenheitstrasse 6, Bremen, 28359 Germany.
[2]University of Bremen, Bibliothekstrasse 1, 28359, Bremen, Germany.
[3]Institute of Marine Science, University of Dar es Salaam, Mizingani Rd, Zanzibar, Tanzania.

*Correspondence to*: E. Fay Belshe (fbelshe@gmail.com)

**Abstract.** The aim of this work was to explore the feasibility of using plant functional traits to identify differences in sediment organic carbon (OC) storage within seagrass meadows. At 19 sites within three seagrass meadows in the coastal waters of Zanzibar, Tanzania, species cover was estimated along with three community traits hypothesized to influence sediment OC storage (above and belowground biomass, seagrass tissue nitrogen content, and shoot density). Sediments within four biogeographic zones (fore reef, reef flat, tidal channel, and seagrass meadow) of the landscape were characterized, and sediment cores were collected within seagrass meadows to quantify OC storage in the top 25 cm and top meter of the sediment. We identified five distinct seagrass communities that had notable differences in the plant traits, which were all residing within a thin veneer (ranging from 19 to 78 cm think) of poorly sorted, medium-coarse grained carbonate sands on top of carbonate rock. One community (B), dominated by *Thalassodendron ciliatum,* contained high amounts of above ($972\pm74$ g DWm$^{-2}$) and belowground ($682\pm392$ g DWm$^{-2}$) biomass composed of low elemental quality tissues (leaf C:N=24.5; rhizome C:N=97). While another community (C), dominated by small-bodied ephemeral seagrass species, had significantly higher shoot density (4178 shoots m$^{-2}$). However, these traits did not translate into differences in sediment OC storage and across all communities the percentage of OC within sediments was similar and low (ranging from 0.15% to 0.75%), as was the estimated OC storage in the top 25 cm ($14.1\pm2.2$ Mg C ha$^{-1}$) and top meter ($33.9\pm7.7$ Mg C ha$^{-1}$) of sediment. These stock estimates are considerably lower than the global average ($194.2\pm20.2$ Mg C ha$^{-1}$) reported for other seagrass ecosystems, and on the lower end of the range of estimates reported for the Tropical Indo-Pacific bioregion (1.9 to 293 Mg C ha$^{-1}$). The uniformly low OC storage across communities, despite large inputs of low-quality belowground tissues in community B, indicate that the geophysical conditions of the coarse, shallow sediments at our sites were not conducive to OC stabilization, and outweighed any variation in the quantity or quality of seagrass litter inputs. These results add to growing body of evidence showing that geophysical conditions of the sediment modulate the importance of plant traits in regards to retention of OC within blue carbon ecosystems, and cautions against the use of plant traits as a proxy for sediment OC storage across all seagrass ecosystems.

# 1 Introduction

Seagrasses influence key ecological functions within coastal ecosystems through their productivity and by trapping sediment, altering hydrodynamics, and modifying biogeochemical processes in the water column and sediment (Duarte and Chiscano, 1999; Marba et al., 2006). Through their effects on ecosystem processes seagrasses provide numerous ecosystem services including sediment stabilization, coastline protection, nutrient cycling, pathogen reduction, support of fisheries, and enhancement of biodiversity (Duffy, 2006; la Torre Castro and Rönnbäck, 2004; Lamb et al., 2017; Orth et al., 2006). In the last two decades, seagrasses have been recognized as important 'blue' carbon (organic carbon sequestered by vegetated coastal ecosystems) sinks, adding climate regulation to their list of well-established ecosystem services (Duarte et al., 2005; Fourqurean et al., 2012a; Macreadie et al., 2014; Mateo et al., 1997; Nellemann et al., 2009; Pergent et al., 1994; Romero et al., 1994).

A surge in research efforts has revealed the wide range (up to 18-fold) of sediment organic carbon (OC) storage within seagrass sediments, with OC stocks varying with seagrass species (Gullström et al., 2017; Lavery et al., 2013; Serrano et al., 2016a; 2014), plant characteristics (Dahl et al., 2016; Jankowska et al., 2016; Samper-Villarreal et al., 2016), meadow attributes (Armitage and Fourqurean, 2016; Samper-Villarreal et al., 2016; Serrano et al., 2014; 2016b), sediment characteristics (Campbell et al., 2014; Dahl et al., 2016; Miyajima et al., 2017; Röhr et al., 2016; Serrano et al., 2016a), landscape configurations (Campbell et al., 2014; Gullström et al., 2017; Lavery et al., 2013; Phang et al., 2015) and climatic zones (Fourqurean et al., 2012a; Lavery et al., 2013; Miyajima et al., 2015). The potential for high variability in OC stocks presents a formable obstacle for reliably valuing the ecosystem service of OC storage because baseline stock estimates are needed before conservation or restoration can be incentivized under a blue-carbon framework (Barbier et al., 2011; Costanza et al., 1997; 2014; Herr et al., 2012; Macreadie et al., 2014). To achieve IPCC tier 3 standards of accuracy for OC stock inventories considerable sampling effort is required (Howard et al., 2014; Macreadie et al., 2014).

A potential solution is to utilize easy-to-measure functional traits that can be linked to ecosystem functions underlying the service of OC storage as a proxy for sediment OC content (de Bello et al., 2010; de Chazal et al., 2008; Grime, 2001; Kremen, 2005). Plant functional traits have been shown to be valuable tools for assessing and managing ecosystem services (de Bello et al., 2010; Díaz et al., 2007). An important trade-off of plant traits linked to OC cycling, known as the 'fast-slow plant economic spectrum', consists of a suite of coordinated characteristics that either promote fast carbon acquisition and decomposition, or promote the conservation of resources within well-protected tissues with inherently slower decomposition rates (Conti and Díaz, 2012; Díaz et al., 2004; Freschet et al., 2012; Grime, 2001; Reich et al., 1997; Wright et al., 2004). Acquisition traits such as high specific leaf area, high nutrient content, low tissue longevity and density are consistently associated with high OC inputs via photosynthesis and high OC losses through decomposition (Cornwell et al., 2008; Díaz et al., 2004; Grime et al., 1997; Herms and Mattson, 1992; Reich, 2014; Wright et al., 2004). Conservative traits include the opposite of the above characteristics and promote slow-growing, long-lived biomass with low OC losses

via decomposition. At the ecosystem level, acquisition traits promote high carbon fluxes, while conservation traits are conducive for the retention of OC stocks (Conti and Díaz, 2012; De Deyn et al., 2008; Díaz et al., 2009; Wardle et al., 2004).

Several seagrass traits have been proposed to be influential on OC sequestration and storage within seagrass sediments (Duarte et al., 2011). Canopy characteristics, such as high leaf density and complexity, have been shown to enhance the OC burial flux into the sediment by filtering and trapping particles from the water column and promoting sediment deposition and retention (Peterson et al., 2004; Duarte and Chiscano, 1999; Duarte et al., 2005; Gacia and Duarte, 2001; Gacia et al., 1999; Hendriks et al., 2008; Samper-Villarreal et al., 2016); and, seagrass sediment OC stocks have been positively correlated with shoot density, both directly (Dahl et al., 2016) and indirectly (Serrano et al., 2014). Seagrass tissue stoichiometry has been correlated with decomposition rates, with tissues containing relatively higher nitrogen and phosphorus content decomposing faster (Enriquez et al., 1993), at least in the initial phase of decomposition (Berg and McClaugherty, 2014). Low concentrations of nitrogen (C:N ratio above 20-25) within tissues indicate the potential for microbial nitrogen limitation, necessitating nitrogen immobilization from the environment and resulting in low carbon-use efficiency during litter decomposition (Berg and McClaugherty, 2014; Hessen et al., 2004; Sinsabaugh et al., 2013). Furthermore, the nutrient content of tissues co-vary with other structural and chemical properties that reflect the plant species' ecological strategy, and can serve as a proxy of tissue quality and decomposability (Birouste et al., 2012; Cornwell et al., 2008; Freschet et al., 2012; Zechmeister-Boltenstern et al., 2015). Seagrass biomass has been positively correlated to OC storage (Armitage and Fourqurean, 2016; Gullström et al., 2017; Serrano et al., 2016a). Belowground production of seagrass roots and rhizomes places OC directly into sediments, which can be stabilized on mineral surfaces, within aggregates, or if microbial activity is suppressed due to lack of oxygen (Belshe et al., 2017; Duarte et al., 2011). In addition, the binding of the sediment by the root-rhizome system (Christianen et al., 2013) and the high lignin content of belowground tissues (Klap et al., 2000), can promote OC storage. And larger plants disproportionately contribute to OC accumulation by shedding more biomass per unit ground area (Garnier et al., 2004; Lavorel and Grigulis, 2011). Seagrass interspecies variation in these traits place them within the continuum of the 'fast-slow' plant economic spectrum, with small-bodied, ephemeral species, such as *Halophila* spp., *Halodule* spp., and *Zostera* spp. on the 'fast' acquisition end, and large-bodied, persistent species, such as *Enhalus* spp., *Thalassia spp.* and *Posidonia* spp., on the 'slow' conservation end (Orth et al., 2006).

The aim of this study was to determine whether seagrass community traits can be linked differences in sediment OC content within meadows residing in the open coastal waters of Zanzibar, Tanzania. Our sites were located within three meadows that contained up to eight co-occurring seagrass species, with a wide breath of functional traits (Gullström et al. 2002), all residing within a landscape with similar abiotic conditions (Shaghude et al., 2002). Our goal was to add to the growing body of evidence investigating where, and to what extent, plant community traits can be used to determine the size and variability of OC storage within seagrass sediments.

## 2 Methods

### 2.1 Description of study sites

This study was conducted within the Tropical Indo-Pacific bioregion (Short et al., 2007) in diverse seagrass meadows of the Western Indian Ocean (WIO), specifically in coastal waters of Unguja Island (-6.15809°S, 39.19181°E) of Zanzibar, Tanzania. The climate is tropical, with temperatures between 27 and 35°C, with an annual rainfall of 1600 mm that is strongly influenced by two opposing monsoon seasons driven by the southeast monsoonal circulation of the central WIO (Mahongo and Shaghude, 2014; McClanahan, 1988). The northeast monsoon occurs from November to February and the southeast monsoon occurs from April to September. The regional hydrodynamics are complex and primarily influenced by ebb-flood tidal phases but are also influenced by the East African Coastal Current (EACC) and monsoon winds (Mahongo and Shaghude, 2014; Shaghude et al., 2002; Zavala-Garay et al., 2015). The tidal cycles are semi-diurnal ranging from mesotidal during neap tide (~1 m amplitude) to macrotidal (from 3 to 4 meters in amplitude) during spring tide (Shaghude et al., 2002; Zavala-Garay et al., 2015). Strong tidal currents can reach velocities that range from 0.25 to 2 m/s (Shaghude et al., 2002).

Sample sites were established within three seagrass meadows (M1, M2, M3) in open coastal waters adjacent to coral cays west of the main city, Zanzibar Town (Figure 1). These meadows were chosen because they contained a range of seagrass species (up to eight species) with different life-history strategies, and at the same time had similar landscape positions and abiotic properties, such as shallow water depth, carbonate sediments, and negligible terrestrial inputs (Shaghude et al., 2002). M1 is located in shallow waters (70 cm – 380 cm in depth) to the southeast of Kibandiko Island and encompasses an area of 15 hectares, which include several small intermittent patch reefs. M2 is also located 1.5 km to the west of M1, and encompasses an area of 4.8 hectares. M2 resides within a shallow lagoon (50 cm – 320 cm in depth) adjacent to a sand spit and fringing reef on the north-eastern side Changu Island. M3 covers 4.6 hectares and is located in shallow waters (50 cm – 375 cm in depth) north of Chumbe Island, adjacent to patch reefs and a sand spit. M3 resides 16 to 17 km south of M1 and M2, respectively. The seagrass within these meadows are growing within a shallow sediment layer on top of an uplifted Pleistocene carbonate platform (Kent et al., 1971; Short et al., 2007). The sediments are biogenic, with the major constituents being benthic foraminifera, molluscs (pelecypods and gastropods) and coral, with negligible terrigenous inputs (G.R. Narayan unplub.; Shaghude et al., 2002).

During October 2013, physical properties (temperature, pH, dissolved oxygen and conductivity) of the water column were measured using a WTW 3430 multi-parameter probe (Weilheim, Germany) within the three meadows. Light levels at the surface (I0) and bottom (Id) of the water column were measured with Li-1400 (Li-Cor Biosciences, Lincoln, Nebraska, USA). Light attenuation (k) at a given depth (d) was calculated using the following equation: $k = \ln(I_d/I_0)/\text{-}d$. To characterize the sediments of the landscape, sediments were collected within four biogeographic zones (reef flat, fore reef, tidal channel and seagrass meadow). For this landscape-level sediment characterization, the upper 5-10 cm of sediment was collected using a Van Veen sampler (3 mm plate, 250 cm²) at 29 locations following the bathymetric gradient and spatially

distributed to cover the four biogeographic zones. Sediment samples were rinsed with clean freshwater in order to remove soluble components and dried at 40°C for at least 48h. Two subsamples (of each set) were sieved in a stack-shaker sieve for 10 min. To assess differences in local sediment characteristics, compared to landscape sediment properties, surface sediments within different seagrass communities (see below for details on communities) were collected by scooping the top 2-3 cm of sediment by hand into a 50 ml falcon tube.

## 2.2 Seagrass community composition

Between September 17th and October 17th of 2013, 19 sample sites were established across the three meadows (M1, M2, M3) to capture the zonation of species assemblages found across the extent of each meadow. A snorkeling survey was conducted at each meadow, consisting of five 50 m transects (perpendicular to the coast line) throughout each meadow. Based on this initial survey, six to seven distinct vegetation zones were identified for each meadow. The pattern of zonation within the meadows was a mosaic of patches, following both the depth gradient and running parallel to the coastline. Within each zone, a 0.25 m$^2$ quadrat was haphazardly tossed to establish the specific site locations. Within a meadow, sites ranged from 15 to 370 meters apart, with the average distance of between sites of 261±194 meters for M1, 170±93 meters for M2 and 165±98 meters for M3 (Figure 1).

At each site, seagrass species composition was quantified within six haphazardly tossed 0.25 m$^2$ quadrats by visually estimating seagrass cover and assigned values based on a modified Braun-Blanquet scale (Mueller-Dombios and Ellenberg, 2012). In total, eight seagrass species were identified: *Thalassodendron ciliatum, Cymodocea serrulata, Cymodocea rotundata, Thalassia hemprichii, Syringodium isoetifolium, Halodule univervis, Holophila ovalis,* and *Halophila stipulacea.*

## 2.3 Seagrass functional traits

To quantify traits of each seagrass community, three biomass cores and five ramets of each seagrass species present were collected at each of the 19 sites. Biomass cores encompassing both seagrass shoots and the entire rhizosphere (ranging from 10-30 cm depth) were taken by placing a 13-cm diameter PVC ring on top of the sediment and using a knife and garden trowel to remove all plant biomass within the ring. This methodology was utilized because of the coarse, shallow carbonate sediments at our sites. Plant material was placed directly into a mesh bag (2 mm mesh size), rinsed free of sediment in the field, stored in plastic bags, and frozen for subsequent analysis. After thawing in the lab, seagrasses were sorted by species and short shoot density (shoot m$^{-2}$) was calculated. Green leaves (above-ground biomass) and living root, rhizome, and short-shoots (below-ground biomass) were separated, scraped of epiphytes, and dried at 60°C until a constant weight was reached, then weighed to obtain above and below ground biomass (g DW m$^{-2}$) for each species. Species weights were then summed for core-level estimates of above and below ground biomass. Five seagrass ramets per species were collected from each site and used to quantify the % nitrogen (N) of leaf and rhizome tissue of each species. A section of rhizome and the second-ranked (from youngest) leaf of each of the five shoots was taken, scraped gently to remove epiphytes, and dried at 60°C for

48 hours. Tissue samples were then homogenized with a mortar and pestle and subsequently measured on an elemental analyzer (Euro EX 3000; EuroVector) to determine the % N and % C of each species at each site, and tissue stoichiometry (C:N ratio) was calculated.

### 2.4 Sediment organic carbon

To determine if carbon storage within sediments varied among different seagrass communities, sediment cores were taken within the five seagrass communities (determined from the multivariate analysis) during October of 2014. Three sediment cores were taken with a hand-driven, 7.6 cm internal diameter corer on SCUBA, within each of the five identified seagrass communities and on bare sediment adjacent to, but outside of the seagrass meadows. Within each community, cores were distributed among the three meadows, resulting in one core extracted per community per meadow. Due to the shallow and

variable sediment accumulation on top of the carbonate platform at our sites, the depth of penetration of sediment cores varied from 19 to 78 cm. The presence of the impenetrable carbonate layer was verified manually after the core was extracted by hand or by inserting a metal rod. Core compaction was not measured in this study but was assumed to be minimal due to the coarse sediment composition. We also assumed that there were no historic differences in community composition, plant traits, or meadow extent during past carbon deposition because there were no historic data available at our

sites, which is a limitation of this study.

In the lab, cores were sectioned into 3 cm slices. From each slice, a 15 cm$^3$ (3 x 2.5 x 2 cm) rectangular cavity was used to further subset each slice of sediment, which was subsequently oven dried (60°C) and weighed for dry bulk density determination. Dried sediments were homogenized in a ball mill and % organic carbon (OC) was determined, after acidification with 1 M HCL to remove carbonates, on an elemental analyzer (Euro EX 3000; EuroVector). The OC content

(CC; g C/cm$^3$) of each 3 cm slice was calculated from measured % OC and the dry bulk density (DBD) of the slice following Eq. (1):

$$CC_{slice} = z_{slice} \text{ x } DBD_{slice} \text{ x } OC_{slice} / 100 \tag{1}$$

where $z_{slice}$ is the slice thickness (cm), and the % OC content of the slice is multiplied by 100 to convert % to grams OC per dry weight. The amount of carbon stored in each core was calculated by summing the OC content in each depth increment

(slice). Because the total core length varied among sites (from 19 to 78 cm) total core carbon storage was estimated in two ways. First, estimates of storage in the top 25 cm of sediment were calculated because at this depth there were nearly full data sets in all cores (16 out of 18 cores were longer than 25 cm). Second, to make estimates comparable to other studies, storage in the top meter of sediment was estimated by gap filling missing data down to one meter using a negative exponential model with the drc package (version 3.0-1; Ritz et al., 2015).

### 2.5 Statistical Analysis

To characterize the sediments of the four landscape zones (fore reef, reef flat, tidal channel, seagrass), we applied the Udden-Wentworth scale (Wentworth, 1922) as following: gravel (>2000 μm), coarse sand (1000-2000 μm), medium

sand (500-1000 μm), medium-fine sand (250-500 μm), fine sand (125-250 μm), very fine sand (63-125 μm) and silt (<63 μm). Each individual fraction was calculated as weight percentage of the total bulk sediment. We used the logarithmic Folk and Ward method (Folk and Ward, 1957) to compare sediment gain size distributions because it places more weight on the central portion of the distribution and less on the tails, and was more appropriate for our sediments, which had a large

particle size range (Blott and Pye, 2001). The physical description of sediments was based on the granulometric output and appearance of the bulk sediment after Folk (Folk, 1954). Summary statistics (mean, median (D50), standard deviation, skewness, and kurtosis) were estimated for each zone based on log-transformed data using the G2Sd R package (Fournier et al., 2014). As an indication of variability of grain sizes found within each zone, a measure of the spread of grain sizes (D10-D90) was calculated by subtracting the grain size at which 10% (D10) of the grains are more coarse from the grain size

(D90) where 90% of the grains are found to be more coarse (Blott and Pye, 2001). Differences among the four landscape zones were compared for each grain size class of Udden-Wentworth scale using a Kruskal-Wallis test with a post-hoc t-test with pooled standard deviation. To assess differences in local sediment characteristics, compared to landscape sediment properties, a representative sediment sample of each seagrass community was photographed at high-resolution over a 5 mm grid and qualitatively compared based on appearance and texture (Folk, 1954).

15       Multivariate analyses were used to describe and categorize the patterns in seagrass species assemblages found at the 19 sample sites. First, Braun-Blanquet cover categories were converted to the midpoint of the cover range (Wilkum and Shanholtzer, 1978), square root transformed to down weight the influence of abundant species, and relativized to the total abundance of each site. A Bray-Curtis similarity index was then calculated based on the similarity of species composition and cover among sites (Bray and Curtis, 1957). Then, based on this similarity matrix, both non-metric multidimensional

scaling (NMDS) and hierarchical cluster analysis (average linkage) were preformed to group sites by similarity in seagrass species composition and cover (ter Braak, 1995; Kent and Coker, 1992; Legendre and Legendre, 1998). These categorizations were used to identify the seagrass species assemblages (communities) present in the sampled meadows. The vegan package (version 2.2-0; (Oksanen et al., 2014) in R (R Core Development Team 2016) was used for all multivariate analysis.

25       For all analysis of trait differences among communities, the unbalanced sample design created from the unequal grouping of the original 19 sites into communities (based on similarity of species and cover), necessitated special attention in regards to model appropriateness and validation of assumptions. Model residuals were tested for homogeneity of variance and normality with Levene's and Shapiro-Wilks tests, respectively. Selected models were also validated visually with plots of model residuals (fitted values vs absolute residuals (homogeneity of variance), a qqplot comparing the distribution of the

standardized residuals to the normal distribution (normality), and a lag plot of the raw residuals vs the previous residual (independence; Zuur et al. 2009). Further, spatial independence was confirmed with variogram plots of model residuals using the gstat package (Zuur et al., 2009).

Differences in above and below ground biomass among communities and meadows were determined using a two-way ANOVA with post-hoc Tukey HSD at $p \leq 0.05$ significance level on log transformed data to meet model assumptions.

With the transformation, both assumptions of homogeneity of variance (Levene's test, AG: $F$=1.262, $p$=0.274; BG $F$=0.609, $p$=0.833) and normality (Shapiro-Wilk, AG: $W$=0.969, $p$=0.180; BG: $W$=0.961, $p$=0.126) were met. Because sites were distributed across three meadows, models included both community and meadow as direct effects along with their interaction. All models were fit using the base package in R (R Core Development Team 2016).

Differences in short-shoot density among the seagrass communities and meadows were determined using a generalized linear model, specifically a negative binomial model (link=log) because the data were counts and found to be over dispersed (Zuur et al., 2009). The negative binomial distribution allows for variances not equal to the mean and does not necessitate equal variances among groups (Zuur et al., 2009). Differences among communities, and across meadows within each community, were determined when there was no overlap in 95% confidence intervals of predicted model estimates. All models were fit using the MASS package (version 7.3-35; (Venables and Ripley, 2002) in R.

The % N of each community was estimated by calculating the mean and standard deviation of the % N weighted by the abundance of each species present within the community specific to each meadow. Because of our unequal sample sizes and variance heterogeneity, communities and meadows within a community, were simply compared visually and considered different when there was no overlap between 95% confidence intervals, which were calculated as the weighted mean ± with $t_{0.95}$ *weighted SD, with $t_{0.95}$ =2.26 based on a t-distribution to account for the smallest sample size of our groups (n=10).

To explore how OC varied among communities and across our sites, models that included both community and meadow as direct effects were evaluated. Because of our relatively small sample size (n=18) and the general rule of thumb of needing ~10 data points for each parameter estimated, we did not include the interaction among community and meadow. Models were fit for each response variable (percent OC in the top 25 cm, OC storage to 25 cm, and OC storage to 1 m) using an ANOVA with post-hoc Tukey HSD at $p \leq 0.05$ significance level. Model validation of normality (Shapiro Wilks test) were met for all OC models and variogram plots of model residuals showed no clear patterns indicating that the assumption of independence was met (Supplementary Figure S1). All graphics were produced with the ggplot package (version 1.0.0; Wickham, 2009) in R (R Core Development Team 2016).

## 3 Results

### 3.1 Seagrass meadow environment

Physical properties of the water column were similar among meadows, with pH ranging from 8.19 to 8.31 ($F_{2,35}$=9.01, $p$=0.06), dissolved oxygen ranging from 6.5 to 8.8 mg/L ($F_{2,35}$=2.53, $p$=0.09), conductivity ranging from 53.7 to 54.1 S/m ($F_{2,35}$=0.18, $p$=0.84). Water temperature ranged from a mean of 26.4°C in M1, to 26.3°C in M2 and 27.1°C in M3. Light attenuation (Kd) through the water column was similar among meadows (mean Kd= 0.35, $F_{2,29}$=1.45, $p$=0.25). Sediments across the landscape were composed of coarse to fine sized carbonate sands, that were poorly sorted but actively reworked. There were no major (compositional or granulometric) differences among the four zones, with all classified as poorly-sorted, gravelly sand with negative skewed distributions, indicating a tail of coarser particles (Table 1). All regions contained

approximately 15% gravel, 84% sand, and 1% mud. However, the median grain size within seagrass meadows (D50=641 μm) was slightly smaller, but within the distribution spread (D10-D90=1938 μm), than sediments from the reef flat (865 μm; D10-D90=1937 μm), fore reef (779 μm; D10-D90=1911 μm) or sediments found in deeper areas of the tidal channel (750 μm; D10-D90=1995; Table 1). When comparing zones for each Udden-Wentworth size classes individually, there were no differences among zones in regards to their abundance of gravel (>2000 μm; $H(2)$=1.27, $p$=0.736), medium sand (500-1000 μm; $H(2)$=0.732, $p$=0.866), fine sand (125-250 μm; $H(2)$=1.551, $p$=0.671), very fine sand (63-125 μm; $H(2)$=2.138, $p$=0.544) and silt (<63 μm; $H(2)$=4.345, $p$=0.227; Figure 2). However, there were differences among zones in their abundance of coarse sand (1000-2000 μm; $H(2)$=14.328, $p$=0.003) and medium-fine sand (250-500 μm; $H(2)$=8.071, $p$=0.045), with seagrass sediments containing a lower abundance of coarse sand than the fore reef ($p$=0.0004) and reef flat ($p$=0.0008), and seagrass sediments containing a higher abundance of medium fine sand than the reef flat ($p$=0.013; Figure 2). At the local scale, there were no large qualitative visual differences among surface sediments beneath the different seagrass communities, and all were consistent with the sediment characterization of the region (Supplementary Figure S2).

### 3.2 Seagrass community composition

Five distinct seagrass species assemblages were identified, here-after referred to as communities A, B, C, D, and E (Figure 3). The first two communities, A and B are monospecific, composed 100% of *Cymodocea serrulata* (CS) and *Thalassodendron ciliatum* (TC), respectively (Figure 3). Although a single species does not fit the strict definition of a community, we use the terminology for congruity throughout the manuscript. Community C was comprised mostly of small-bodied, ephemeral species 67% *Halodule uninervis* (HU), 19% *Cymodocea rotundata* (CR), 8% *Halophila ovalis* (HO), 1% *Halophila stipulacea* (HS), but also contained a small percentage (5%) of *Thalassia hemprichii* (TH). Community D was dominated by TH (91%) with a lesser occurrence of CR (8%) and TC (1%). Community E had the highest evenness of all communities with 46% TH, 26% CS, 22% *Syringodium isoetifolium* (SI), 5% HU, and 1% TC.

### 3.3 Community traits

There was a significant effect of both community ($F_{4,41}$=46.45, $p$<0.0001) and meadow ($F_{2,41}$=13.15, $p$<0.0001) on above ground (AG) biomass but no interaction effect ($F_{7,41}$=1.73, $p$=0.128). Community differences were driven by the significantly higher AG biomass in community B (972±74 g DWm$^{-2}$), which contained at least seven-fold higher AG biomass than the other communities and was dominated by the large-bodied species *T. ciliatum* (TC; Figure 4; Supplementary Table S1). There were also significant differences among the other four communities, with community A (127±33 g DWm$^{-2}$) containing greater AG biomass than communities C, D and E; community C (38±18 g DWm$^{-2}$) composed of small-bodied species having the lowest AG biomass, and communities' D (69±50 g DWm$^{-2}$) and E (67±33 g DWm$^{-2}$) with similar intermediate AG biomass (Figure 4). Differences among meadows were driven by M3 (96±52 g DWm$^{-2}$) having significantly less AG biomass than meadows 1 and 2 (M1: 350±72 g DWm$^{-2}$; M2: 123±40 g DWm$^{-2}$), which was

due to the absence of the high-biomass community B within M3. Within communities, there were no significant among-meadow differences in AG biomass (Supplementary Table S1).

Below ground (BG) biomass followed a similar pattern with significant effects of community ($F_{4,41}$=11.01, $p$<0.0001) and meadow ($F_{2,41}$=4.140, $p$=0.023) but no interaction effect ($F_{7,41}$=1.81, $p$=0.111). Community differences were
again mainly driven by a significantly higher BG biomass in the TC-dominated community B (682±392 g DWm$^{-2}$), which contained on average twice as much BG biomass than the other four communities (Figure 4). However, BG biomass within community B was not significantly different from community E (392±144 g DWm$^{-2}$), which had the highest species evenness. BG biomass within community E was not significantly different than community D (303±103 g DWm$^{-2}$), but was significantly greater than Communities A (256±66 g DWm$^{-2}$) and C (233±82 g DWm$^{-2}$), which contained similarly low BG
biomass (Figure 4; Supplementary Table S2). Differences among meadows were due to M2 (325±170 g DWm$^{-2}$) containing on average significantly less BG biomass than meadows 1 and 3 (M1:390±132 g DWm$^{-2}$; M3:335±109 g DWm$^{-2}$). Within communities, there were no significant among-meadow differences in BG biomass (Supplementary Table S2).

There was a significant effect of community on seagrass short shoot density ($\chi^2$=45.1, df=4, $p$<0.001), and a marginally significant effect of meadow ($\chi^2$=5.49, df=2, $p$=0.064). Community C, dominated by small-bodied ephemeral
seagrass species, had the highest shoot density with an estimated mean shoot density of 4178 shoots m$^{-2}$ (based on the negative binomial model); however, only meadows 1 (M1:5285 shoots m$^{-2}$) and 3 (M3: 5696 shoots m$^{-2}$) were found to contain densities different from other communities (determined by no overlap in 95% confidence intervals of model predictions, Figure 5). Meadow 2 of community C (M2: 2105 shoots m$^{-2}$) and the meadow-specific estimated means of the remaining communities were all similar and ranged from 775 to 1781 shoots m$^{-2}$. Within communities A, B, D and E, there
were no consistent trends or significant differences among meadows in predicted short shoot densities (Figure 5).

The nitrogen content within seagrass leaves varied among seagrass communities (determined by no overlap in 95% CI), with community D having the highest % nitrogen (M1:2.58±0.11%, M2: 2.77±0.10%, M3: 2.45±0.28%), which was significantly higher than communities A (M1:1.34±0.08%, M3:1.46±0.06%) and B (M1:1.56±0.05%, M2:1.52±0.04%) regardless of within community variation due to among meadow-specific differences (Figure 6). The higher level of leaf
nitrogen in community D was driven by the high relative abundance of *T. hemprichii*, which contained the highest leaf nitrogen (on average 2.46%) of any seagrass species. Communities C (M1:1.45±0.48%, M2:1.19±0.23%, M3:2.20±0.23%) and E (M1:1.61±0.11%, M2:1.48±0.12%, M3:1.70±0.21%) had intermediate leaf nitrogen concentrations. However, within community C one meadow (M3) contained leaf N content on par with community D due to the presence of *C. rotundata* (2.2%N) at this site. Leaf stoichiometry (C:N ratio) was on average within the range of 20-25, for all communities except
community D (mean C:N =16.9, M1:19.3, M2:13.8, M3:17.6). C:N ratios for communities A (mean= 24.3, M1:25.6, M3:22.9) and B (mean=24.5, M1:23.6, M2:25.4) were at the upper limit of the range measured for seagrass leaves at our sites.

The nitrogen content within seagrass rhizomes did not significantly vary among communities or meadows, with the weighted mean % nitrogen in all communities across all meadows ranging from 0.42% to 0.67% (Figure 6) and rhizome C:N

ratios ranging from 78 to 97. However, within community D there was notably higher variability in rhizome nitrogen content within M2, which had the highest tissue % N (0.92%) due to the high relative abundance (74%) of *Thalasia hemprichii*.

### 3.4 Sediment carbon

The percentage of OC within the sediment was low within all communities (A-E), varying from a maximum of 0.75% in surface sediments to a minimum of 0.15% down core (Figure 7). There were no differences in % OC in the top 25 cm (where all cores had data) among seagrass communities (A-E; $F_{4,9}$=1.34, $p$=0.34) or among meadows ($F_{2,9}$=3.16, $p$=0.09) but there was significantly higher % OC in communities with seagrass (A-E) compared to bare sediment (F, $F_{5,11}$=6.97, $p$=0.004; Supplementary Table S3). Most cores (13 out of 18) exhibited the typical trend of decreasing % OC with depth into the sediment, with the notable exception of two cores taken adjacent to seagrass meadows (F: bare sediment; Figure 7), where % OC increased with depth, which calls into question our assumption that seagrass meadow extent has not changed over time. Sediment bulk density ranged from 0.939 to 1.714 g DW cm$^{-3}$, with mean and median values of 1.303 and 1.299 g DW cm$^{-3}$, respectively (Supplementary Figure S3). Patterns in sediment OC density (g OC cm$^{-3}$) mirrored the trends seen in % OC (Supplementary Figure S4).

The OC stored within the top 25 cm of sediment did not differ among seagrass communities (A-E, $F_{4,9}$=1.43, $p$=0.30) or among meadows ($F_{2,9}$=3.35, $p$=0.08), and was low, on average 14.1±2.2 Mg C ha$^{-1}$, but was significantly higher than adjacent bare sediment (7.5±2.1 Mg C ha$^{-1}$, $F_{5,11}$=7.96, p=0.002; Figure 8; Supplementary Table S3). Similarly, OC stored down to 1 m did not differ among seagrass communities (A-E, $F_{4,9}$=0.20, $p$=0.93) or meadows ($F_{2,9}$=0.04, $p$=0.99) and was on average 33.9±7.7 Mg C ha$^{-1}$, but with the gap filled data there was no longer a significant difference in OC storage between seagrass communities and bare sediment (19.3±8.2 Mg C ha$^{-1}$, $F_{5,11}$=1.448, $p$=0.28).

## 4 Discussion

In three seagrass meadows off the coast of Zanzibar Town, Tanzania, we identified five distinct seagrass communities. Even with the natural variation across meadows there were still notable differences among communities in key plant traits shown to influence ecological processes linked to OC sequestration and storage in other ecosystems (Aerts and Chapin, 2000; Chapin, 2003; Díaz et al., 2004). However, these trait differences did not translate into differences in sediment OC stocks among seagrass communities. The OC storage in the top 25 cm (14.1±2.2 Mg C ha$^{-1}$) or the top 1 m (33.9±7.7 Mg C ha$^{-1}$) of sediment at our sites was comparatively lower than the global average (194.2±20.2 Mg C ha$^{-1}$ in the top 1 m) for seagrass ecosystems (Fourqurean et al., 2012a); however, fell within the range of storage (1.9 to 293 Mg C ha$^{-1}$) reported for seagrass sediments within the Tropical Indo-Pacific bioregion (Alongi et al., 2016; Campbell et al., 2014; Fourqurean et al., 2012a; Miyajima et al., 2015; Phang et al., 2015; Rozaimi et al., 2017; Schile et al., 2016). Compared to other sites within the Indo-Pacific bioregion, OC storage at our sites were lower than stocks reported for meadows in Thailand (37.5 to 120.5 Mg C ha$^{-1}$; Miyajima et al., 2015), Malaysia (46 to 70 Mg C ha$^{-1}$; Rozaimi et al., 2017), Indonesia (34.3 to 293.3 Mg C ha$^{-1}$; Alongi et

al., 2016) and Singapore (129.4 to 149.6 Mg C ha$^{-1}$; Phang et al., 2015). Although, in comparison to sites on the western side of the Indo-Pacific region, our sites' OC stocks fell within the range reported for sites in the Arabian Gulf (1.9 to 109 Mg C ha$^{-1}$; Campbell et al., 2014), and Zanzibar, mainland Tanzania, and Mozambique (21.3 to 73.8 Mg C ha$^{-1}$ in the top 50 cm; Gullström et al., 2017).

A clear contrast emerges when comparing OC storage within community B, dominated by the large-bodied, persistent species *T. ciliatum*, to locations that contain seagrass species with similar life-history strategies and traits. Even with the combined attributes of producing a high quantity (AG biomass: 972±74 g DWm$^{-2}$; BG biomass: 682±392 g DWm$^{-2}$) of low-quality tissues (leaf: 1.54±0.05 %N; rhizome: 0.46±0.2 %N), community B's sediment OC stocks (32.2±7.9 Mg C ha$^{-1}$) were at least 3-fold lower than what has been reported for *Posidonia oceanica* (105 to 829 Mg C ha$^{-1}$), *Thalassia*

*testudinum* (124 to 210 Mg C ha$^{-1}$) and *Amphibolis antarctica* (115 to 335 Mg C ha$^{-1}$) meadows (Fourqurean et al., 2012b; Mateo et al., 1997; Serrano et al., 2012; 2014). All of these species possess traits that place them on the 'slow' conservation-side of the plant economic spectrum associated with higher ecosystem OC storage (Díaz et al., 2004; Orth et al., 2006; Reich, 2014; Wright et al., 2004). The breakdown of the relationship among plant traits and OC storage in this study indicates that other factors may be interacting to control OC deposition and/or stabilization within the sediment.

On one hand, water flow at our sites is energetic with moderate to high current velocities (ranging from 0.25 to 2 ms$^{-1}$; Shaghude et al., 2002), sediments are poorly sorted, and both sediment accumulation and the amount of fine sediments (~1% <63 size fraction) is low (Table 1). These ecosystem properties are characteristic of low-depositional environments and would support the viewpoint that low OC deposition of aboveground autochthonous litter and allochthonous inputs are limiting OC accumulation. However, the high autochthonous inputs of belowground tissues (up to 1074 g DWm$^{-2}$) at our

sites places up to an estimated 386.6 g OC m$^{-2}$ directly into the sediment, providing direct evidence that OC enters the sediment. The C:N ratio (97) of these belowground inputs approach a theoretical threshold (100) where litter decomposition greatly slows due to nutrient limitation of decomposers (Zechmeister-Boltenstern et al., 2015), and if the tissues of *T. ciliatum* are similar to other long-lived seagrass species they contain a high abundance of complex chemical compounds such as lignin (Kaal et al., 2016; Klap et al., 2000; Papenbrock, 2012; Trevathan-Tackett et al., 2017). Low OC storage with

high autochthonous inputs gives greater weight to the argument that OC is not stabilized within the coarse, shallow sediments of our sites, despite the low-quality of seagrass inputs.

These results fit within the emerging framework that the stabilization of OC within soils and sediments is a whole-ecosystem property (Lehmann and Kleber, 2015; Schmidt et al., 2011). This view posits that all organic matter can decay quickly if conditions are right (Gramss et al., 1999; Hamer et al., 2004; Hazen et al., 2010; Wiesenberg et al., 2004).

However, decomposition can be altered by ecosystem properties that impede the microbial access to, or remineralization of, certain molecules (Lehmann and Kleber, 2015; Schmidt et al., 2011). For example, in sediments with low oxygen concentrations, the decomposition of complex, recalcitrant OC can be impeded (due to a lack of electron acceptors or enzyme cofactors that require oxygen), resulting in the selective preservation of 'oxygen-sensitive' OC (Arnarson and Keil, 2007; Burdige, 2007; Burdige and Lerman, 2006; Hedges and Keil, 1995; 1999; Keil and Mayer, 2014). Likewise, sediment

minerology and aggregation can reduce the bioavailability and accessibility of OC to microbes and enzymes (Arnarson and Keil, 2001; 2007; Hedges and Keil, 1999; Keil and Mayer, 2014; Mikutta et al., 2006; Schrumpf et al., 2013; Six et al., 2004; 1998; Sollins et al., 1996; Tisdall and Oades, 1982). Alternatively, in the absence of ecosystem controls, even low-quality, chemically complex compounds such as lignin can be degraded relatively quickly (Dittmar and Lara, 2001). This view shifts plant input quality into an auxiliary role, with the persistence of sediment OC ultimately determined by geophysical properties of the sediment.

The modulation of the role of plant traits in OC storage by sediment properties is seen when comparing our sites on the western coast of Unguja Island, Zanzibar to meadows located on the south and east coast of the island. At these other locations, sediment OC storage is two to three times higher than what was measured in our sites (40.7 to 73.8 Mg C ha$^{-1}$ in the top 50 cm), and is positively correlated to seagrass biomass at the landscape scale, with the largest stocks located in sediments beneath large, persistent species (Gullström et al., 2017). *T. ciliatum* occurs at all locations, and contains similar amounts (AG biomass: 556±200 g DWm$^{-2}$; BG biomass: 983±564 g DWm$^{-2}$) of low elemental quality (AG tissue %N: 1.4±0.1; BG tissues %N: 0.7±0.1) plant tissues (Gullström et al., 2017). What does differ between our sites and these meadows to the south and east are the sediments. The biogenic carbonate sediments that occur on the western side (where our sites occur) differ greatly from the eastern and southern coasts of the Island (Shaghude et al., 1999). The western carbonate sediments are composed of reefal foraminifera, mollusk, echinoderm and coral components and are characterized as coarse gravely sand (Table 1), whereas the eastern and southern sediments are composed primarily of remnants from calcareous green algae (*Halimeda* spp.; Shaghude et al., 1999), which form algal mounds, allowing for greater deposition of fine particles and deeper accumulations of carbonate mud (Kangwe et al., 2012; Muzuka et al., 2005). The narrow range of sediment properties found across the three meadows we sampled leaves us only the ability to piece together trends with data from others' work and speculate that differences in OC storage among regions of the island are due to the disparity in sediment characteristic, since plant traits were similar. Another limitation of this work is that we are unable to identify the exact control(s) within the sediment environment controlling OC stabilization (or lack thereof), though we hypothesize it is linked to oxygen availability and sediment structure (accessibility).

However, this study does add a key piece to the growing body evidence showing that geophysical conditions of the sediment modulate the importance of plant traits in regards to retention of OC within blue carbon ecosystems (Alongi et al., 2016; Armitage and Fourqurean, 2016; Campbell et al., 2014; Dahl et al., 2016; Miyajima et al., 2017; Röhr et al., 2016; Samper-Villarreal et al., 2016; Serrano et al., 2016a). Here we show that once sediments become very coarse and shallow, large inputs of low-quality seagrass OC are not necessarily stabilized against microbial decay. This extends and contrasts previous work from sites without high sediment loading and fine sediments, which show plant traits (biomass, density, and cover) became better predictors for OC storage as sediments become more coarse (Dahl et al., 2016). This increase in explanatory power by plant characteristics as sediments become coarser was also shown for large-bodied, persistent species (*Posidonia* spp. and *Amphibolis* spp.) inhabiting more exposed sites (Serrano et al., 2016a). Sites with the largest stores of OC recorded for seagrass are negligibly correlated with fine sediment content and occur within dense meadows of the long-

lived species *P. oceanica*, which form and persist in stable environments without high sediment inputs (Peirano and Bianchi, 1995; Serrano et al., 2012; Serrano et al. 2016a). However, as the abundance of fine sediments increase, OC storage can be high even in meadows composed of species with "fast" traits, and characteristics of the sediment become better predictors of OC content (Dahl et al., 2016; Lavery et al., 2013; Röhr et al., 2016; Serrano et al., 2016a; van Katwijk et al., 2011). A positive correlation between fine sediment and OC storage has been shown for small-bodied seagrass species at 20 sites across three bioregions (Temperate Southern Ocean, Tropical Indo-Pacific, and Mediterranean; Serrano et al., 2016a). At adjacent estuarine sites in Thailand with a high contribution of terrestrial inputs and fine sediment, a relatively smaller-bodied seagrass (*Cymodocea serrulata*: 120 Mg C ha[-1]) had higher OC storage than the large-bodied, persistent seagrass (*Enhalus acoroides*: 86 Mg C ha[-1]; Miyajima et al., 2015). A similar association between high OC storage and fine sediment was demonstrated across a range of conditions in the Temperate North Atlantic for the small-bodied species, *Zostera marina* (Dahl et al., 2016). Based on the results presented here, in combination with the findings outlined above, we hypothesize the interaction between plant traits and sediment properties is non-linear, with the effect of sediment properties dominating at the extremes of the sediment spectrum. In high depositional environments with an abundance of fine sediments, characteristics of the sediment overshadow the effect of plant traits on OC storage. In moderate depositional areas with coarser sediments, the importance of plant traits increase and meadows with "slow" traits tend to store more OC. And finally, this study shows that once the flow-regime becomes energetic enough to create very coarse sediments and sediment limitation, properties of the sediment can again outweigh plant traits to limit OC storage even under meadows with traits conducive to OC storage.

This study, placed into the context of the growing body of evidence of the large variation in OC storage in seagrass ecosystems (Campbell et al., 2014; Dahl et al., 2016; Lavery et al., 2013; Miyajima et al., 2015; Röhr et al., 2016; Samper-Villarreal et al., 2016; Serrano et al., 2014; 2016a; 2016b), illustrates the complexity of controls and mechanisms that govern OC storage in seagrass sediments. Even within meadows with similar environmental conditions, data on plant traits or carbon sources (as a proxy for OC input quality) cannot alone provide a full picture of the location or magnitude of sediment OC; therefore, we caution against their singular use as proxies for OC storage. Future efforts should focus on quantifying the interactions among properties of OC inputs (quantity and quality) and a suite of geophysical sediment properties, including minerology, structure, and the full range of the grain size distribution. Once these interactions can be quantified, spatial information on sediment parent material (Hartmann and Moosdorf, 2012) and composition can be integrated with data on seagrass characteristics and extent to better model the spatial variability of OC storage within seagrass sediments.

**5 Conclusion**

In this study, we were unable to link variations in plant traits to differences in sediment OC stocks within diverse seagrass meadows off the coast of Zanzibar Town, Tanzania. Geophysical constraints of the sediment outweighed any effects of trait differences on OC stabilization and resulted in low OC storage across all seagrass communities despite high inputs of low-quality OC within some communities. In spite of being constrained within the particular environment,

seagrasses still managed to store twice as much OC as bare sediment. This highlights the importance of seagrass habitats for OC cycling in coastal marine ecosystems; however, further research is needed to identify under which geophysical conditions seagrass traits can be linked to the ecosystem function of OC storage.

**Acknowledgments**

This project was carried out within the framework and through funding provided by the Leibniz Graduate School SUTAS (Sustainable Use of Tropical Aquatic Systems; SAW-2013-ZMT-4) and the Leibniz Centre for Tropical Marine Research (ZMT) based in Bremen, Germany. E. Fay Belshe was supported by the German Academic Exchange Service (DAAD) with funds from the German Federal Ministry of Education and Research (BMBF) and the People Programme (Marie Curie Actions) of the European Union's Seventh Framework Programme (FP7/2007-2013) under REA grant agreement n° 605728

(P.R.I.M.E. – Postdoctoral Researchers International Mobility Experience). D. Hoeijmakers and N. Herran were supported by SUTAS and M. Teichberg was supported by the German Research Foundation (DFG) within the Individual Grants Program, project SEAMAC, TE 1046/3-1. We thank the following people and organizations that supported this work: D Dr. Muhando and Dr. Jidowii at the Institute of Marine Sciences (IMS), University of Dar es Saalam, and Uli Kloiber and team from the Chumbe Island Coral Park Ltd. (CHICOP). Lastly, we would like to thank Dr. Benjamin Bolker (McMaster

University, Canada) for support and advice on modelling procedures and statistical analysis.

**Competing Interest**

The authors declare that they have no conflict of interest.

**Author Contribution**

All authors contributed to this manuscript; specifically, EF Belshe contributed to the design, data acquisition, analysis,

interpretation and wrote the first draft of the manuscript; D Hoeijmakers contributed to the design, data acquisition and analysis, and provided critical review of the manuscript; N Herran contributed to the design, data acquisition, analysis and provided critical review of the manuscript, M Mtolera contributed to the design and data acquisition, and critically revised the manuscript; M Teichberg contributed to the design, data interpretation and critically revised the manuscript. All authors have read and approved the final version of the manuscript.

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

**Table 1. Landscape sediment characterization of the four biogeographic zones. Summary statistics (mean, standard deviation, skewness, kurtosis) are based on the logarithmic Folk and Ward method are shown in both μm and phi scale. The distribution spread (D10-D90) of grain sizes is calculated as the difference between the grain size (D10) where 10% of grains are coarser and the grain size (D90) where 90% if the grains are coarser. D50 is the median grain size.**

| Zone | Reef Flat | Fore Reef | Tidal Channel | Seagrass |
|---|---|---|---|---|
| n | 7 | 10 | 4 | 8 |
| Mean grain size (μm) | 2819 | 2352 | 2546 | 1953 |
| SD grain size (μm) | 2.15 | 2.19 | 2.54 | 2.39 |
| Mean grain size (φ) | 0.54 | 0.63 | 0.43 | 0.66 |
| SD grain size (φ) | 1.11 | 1.13 | 1.35 | 1.25 |
| Skewness (φ) | -0.379 | -0.295 | -0.107 | -0.052 |
| Kurtosis (φ) | 1.137 | 1.025 | 0.987 | 0.965 |
| Sorting | Poorly Sorted | Poorly Sorted | Poorly Sorted | Poorly Sorted |
| Texture | Gravelly Sand | Gravelly Sand | Gravelly Sand | Gravelly Sand |
| % Gravel | 15.4 | 13.6 | 16.8 | 14.8 |
| % Sand | 84.1 | 85.6 | 82.1 | 84.4 |
| % Mud (>63 μm) | 0.5 | 0.8 | 1.1 | 0.7 |
| D10 (μm) | 2164 | 2123 | 2189 | 2130 |
| D50 (μm) | 865 | 779 | 750 | 641 |
| D90 (μm) | 227 | 212 | 194 | 192 |
| D10-D90 (μm) | 1937 | 1911 | 1995 | 1938 |

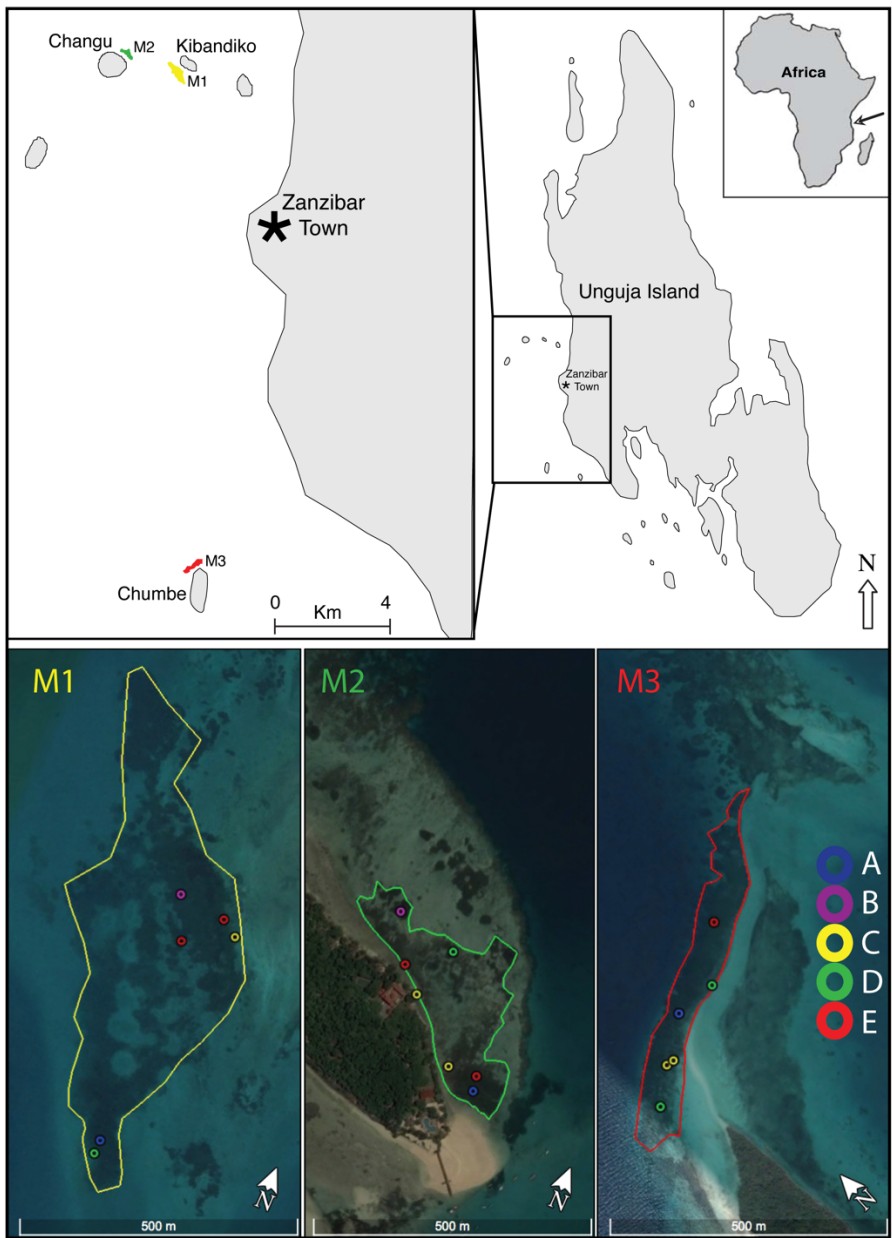

**Figure 1: Study sites were located within three meadows (M1, M2, M3) in open coastal waters adjacent to coral cays west of the main city, Zanzibar Town, Unguja Island (-6.15809°S, 39.19181°E) of Zanzibar, Tanzania. Locations of the 19 sample sites within the three meadows (M1, M2, M3) shown in the bottom panels. Images were produced in Google earth Pro V 7.3.0.3832 (August 18, 2017), Data SIO, NOAA, US Navy, IGA and GEBCO. Images © 2017 DigitalGlobe. http://www.earth.google.com**

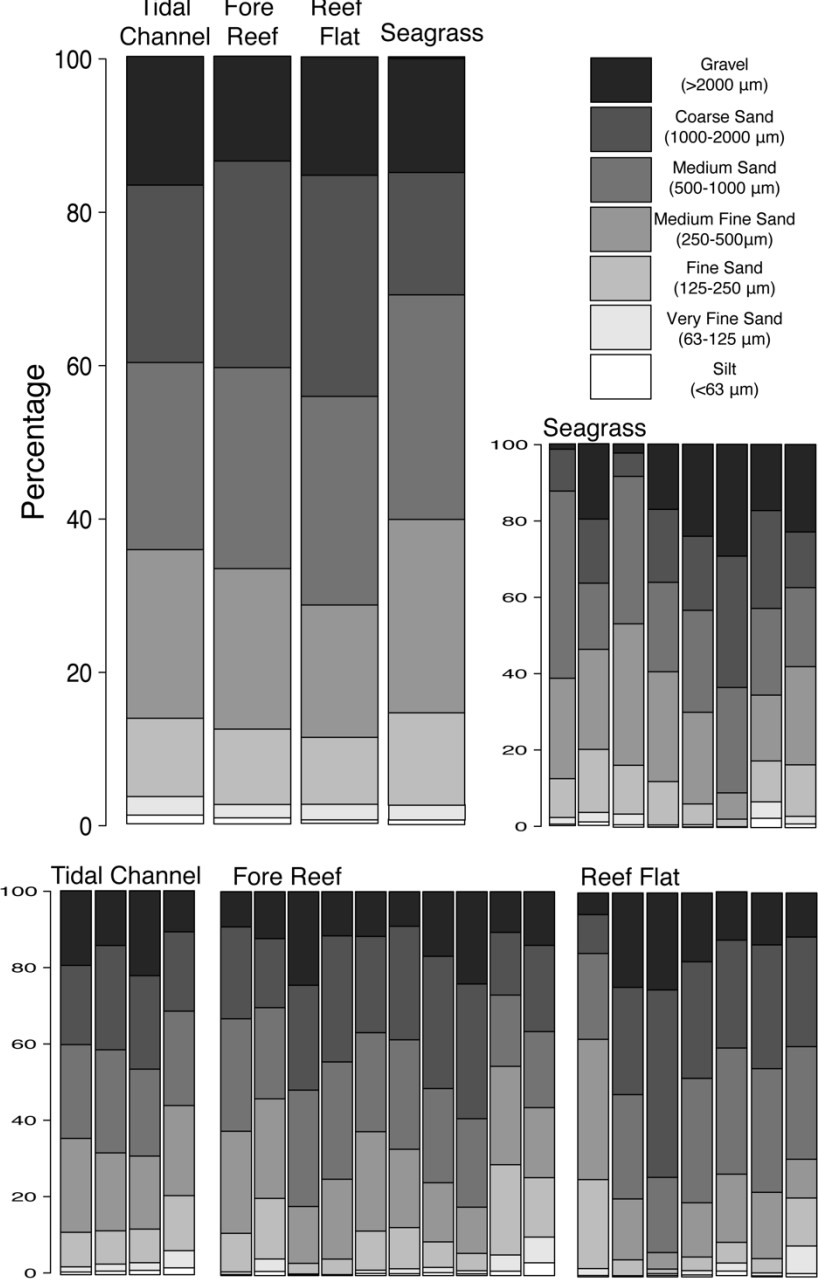

**Figure 2: Average percentage of sediments within each sediment grains size class (top left) for the four biogeographic zones (tidal channel, fore reef, reef flat, and seagrass meadow) found at our study sites. Within each zone, the grain size distributions of each sample are also shown (bottom and left). Grain size classes are based on the Udden-Wentworth grade scale.**

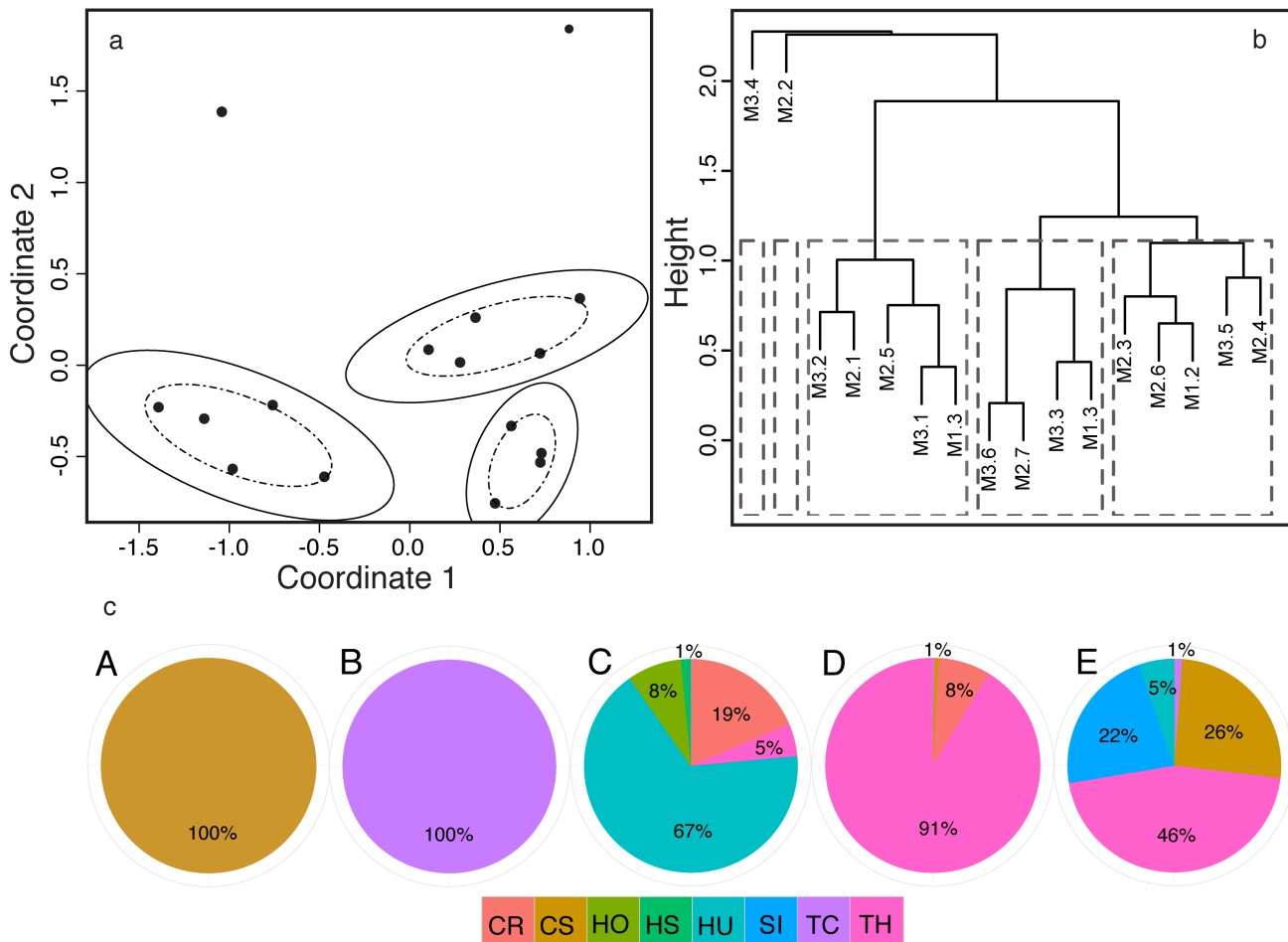

**Figure 3: Seagrass communities were determined by grouping the 19 sites from three meadows (M1, M2, M3) based their similarity in seagrass species composition and cover using two methods. First, a) NMDS ordination plot was used to group sites (●) into a community if they fell within the dashed and solid ellipses representing the 66% and 95% confidence intervals of groupings. Note that within two communities there was 100% overlap in site similarities, so multiple sites are overlain and confidence ellipses were not plotted. Second, b) hierarchical cluster analysis converged on the same five communities. Again, because of the high similarity (100%) of sites within the first two grouping, only one site name is plotted even though multiple sites were grouped into these communities. c) Pie charts of each community (A-E) show the mean percent cover and species composition, with colors representing the different species: CR:** *Cymodocea rotundata*, **CS:** *Cymodocea serrulata*, **HO:** *Holophila ovalis*, **HS:** *Halophila stipulacea*, **HU:** *Halodule univervis*, **SI:** *Syringodium isoetifolium*, **TC:** *Thalassodendron ciliatum*, **TH:** *Thalassia hemprichii.*

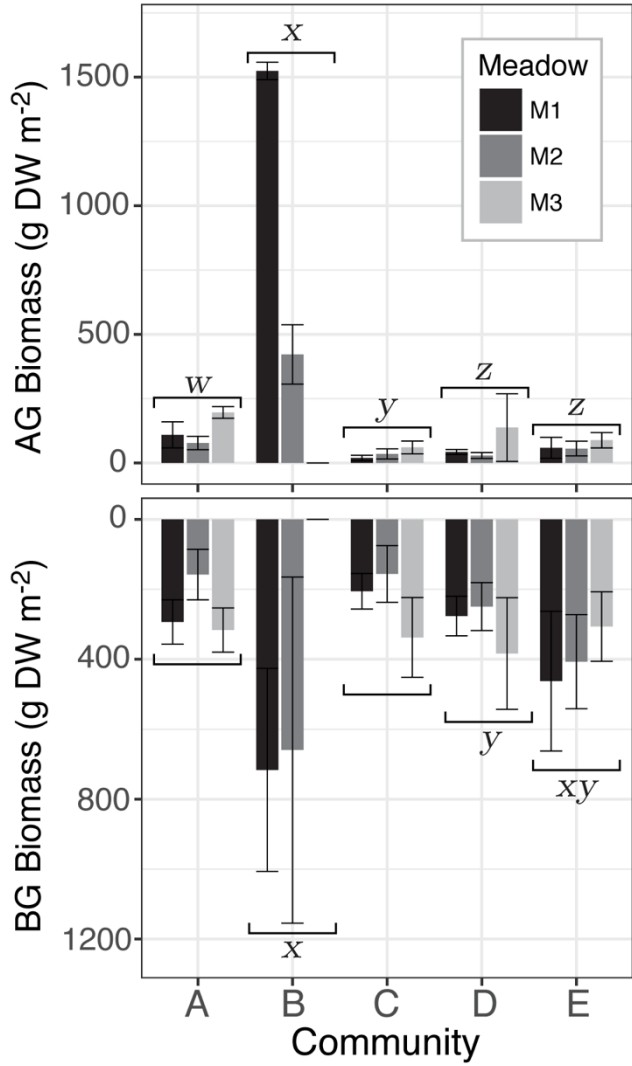

**Figure 4: Mean seagrass above- (AG) and belowground (BG) biomass (g DW m-2) of sites from the three meadows (M1, M2, M3) for each seagrass community (A-E). Whiskers represent the standard deviation and statistical differences among communities at the significance level p ≤ 0.05 indicated by letters (w,x,y,z). Differences among meadows within each community were tested for with Tukey's post hoc test but no significant differences were found (Supplementary Tables S2 and S3).**

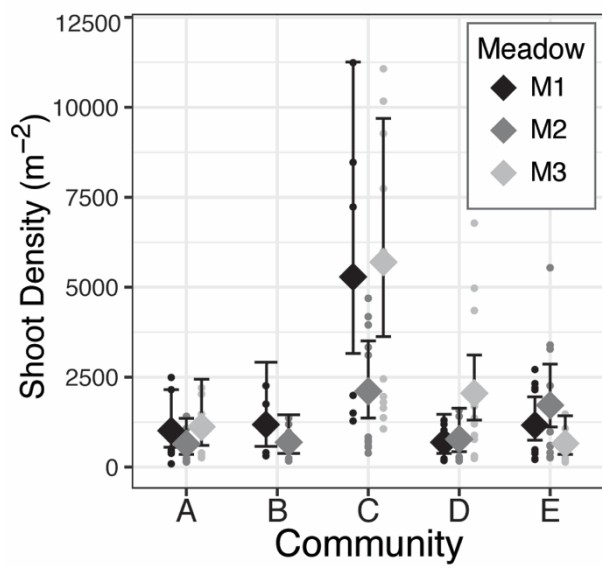

**Figure 5: Meadow-specific estimated mean (◆) and 95% confidence intervals of seagrass shoot density (shoots/m-2) for each seagrass community (A-E).**

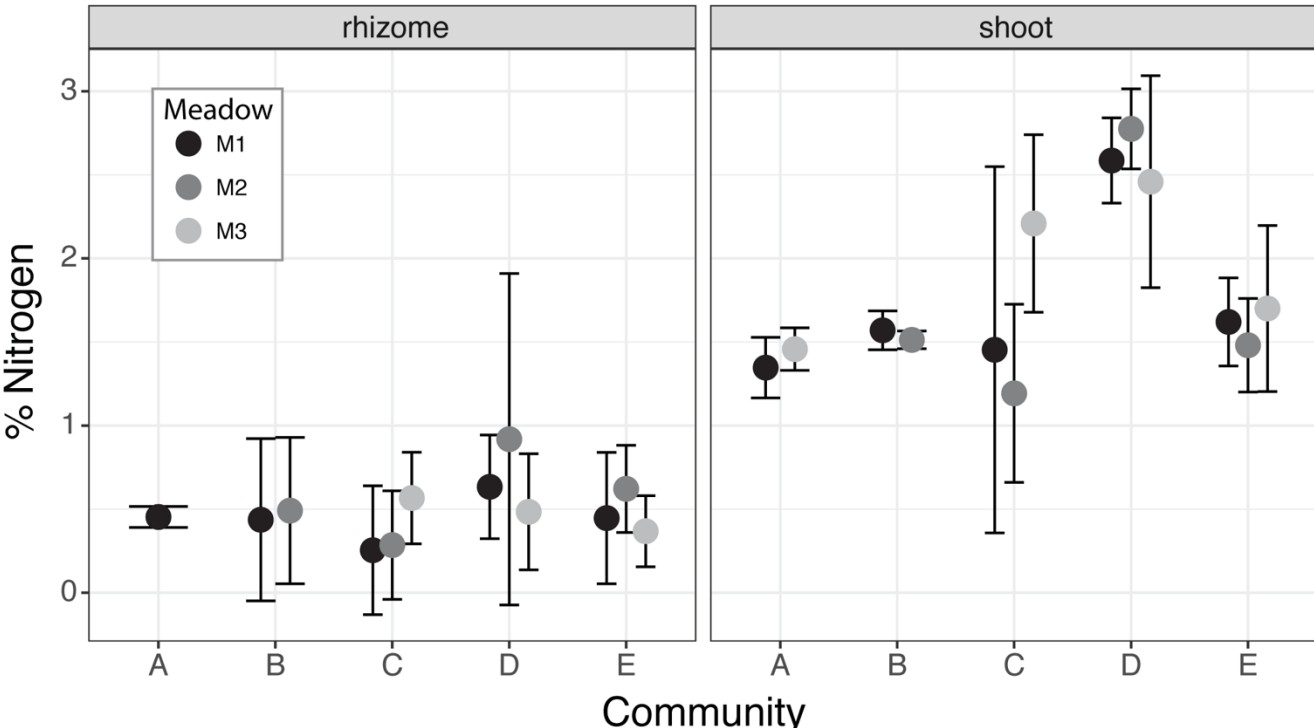

**Figure 6: Percent nitrogen (N) in rhizome (left) and leaf (right) tissues from seagrass species assemblages of each community specific to each meadow (M1, M2, M3). Circles (●) denote the mean and whiskers the 95% confidence intervals of the % N weighted by the meadow-specific abundance of each species present within the community (weighted mean ± $t_{0.95}$*weighted standard deviation).**

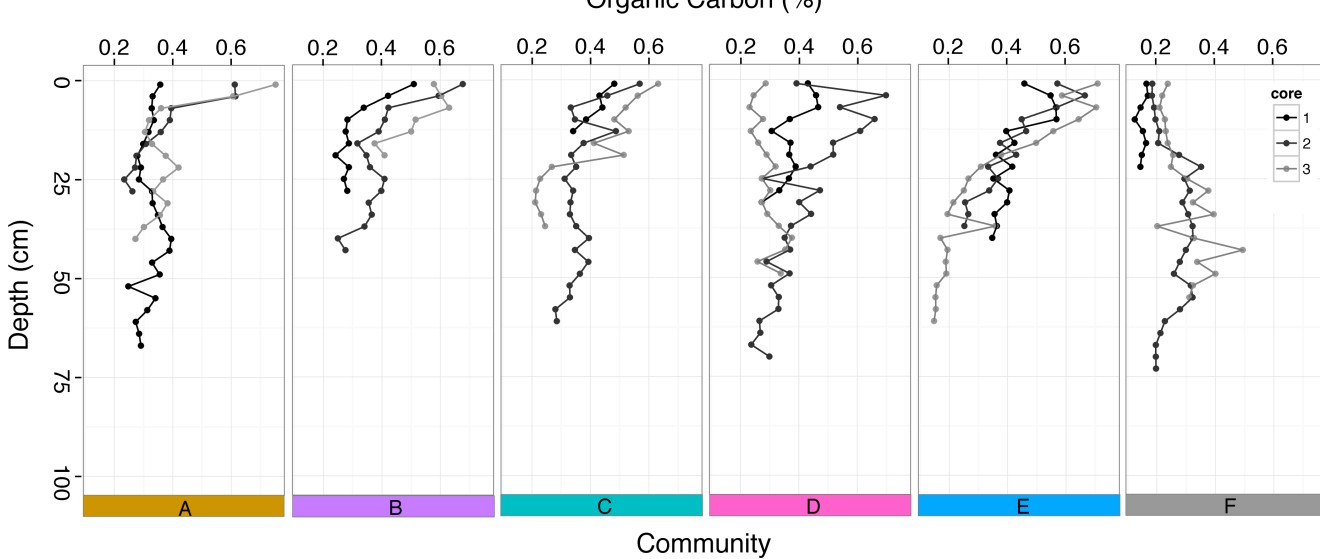

**Figure 7: Percent organic carbon at different depths (cm) down each sediment core taken within the five seagrass communities (A-E) and bare sediment (F).**

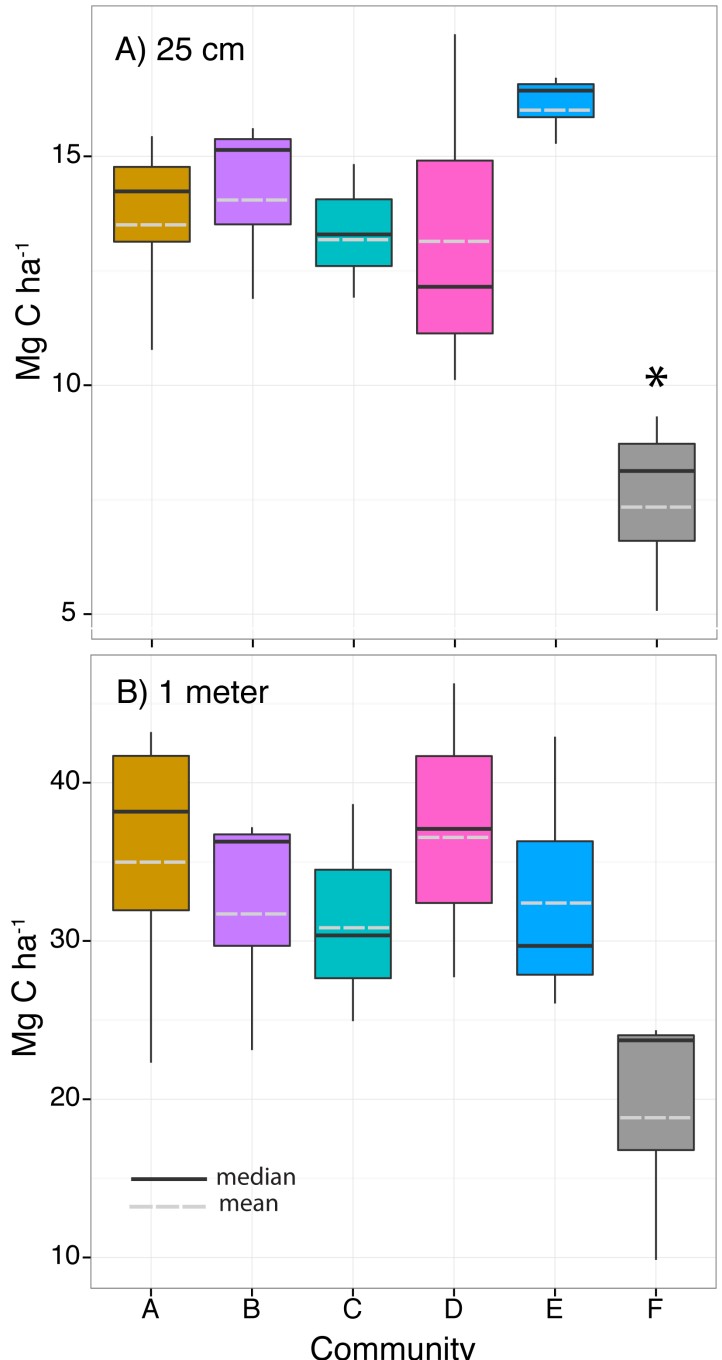

**Figure 8: Organic carbon storage of A) the top 25 cm of the sediment and B) the top meter of sediment within the five seagrass communities (A-E) and bare sediment (F). Box and whiskers denote the 25th and 75th, and the 5th and 95th quartiles, respectively, with the solid line denoting the median and dashed line denoting the mean. Statistical difference at the significance level p ≤ 0.05 indicated by the asterisk (\*).**

