# Peer review of "Seagrass community-level controls over organic carbon storage are constrained by geophysical attributes within meadows of Zanzibar, Tanzania."

_Biogeosciences, 2017_

## Referee Comment (RC1) · Anonymous Referee #1 · 24 Jan 2018

General comment: This manuscript presents findings from a field study of the variation in organic carbon (OC) in seagrass sediments at 19 sites in Tanzania. It hypothesized that three plant traits – biomass (above and below ground), shoot density and N content – might explain seagrass sediment OC content. There was no link found between seagrass sediment OC content and the three functional traits analysed, despite variations in functional traits among seagrass communities. This a finding that would be useful to publish, however the manuscript needs to be improved. The introduction needs to be further developed and references updated. The methodology needs to be

more detailed, as it stands it is not really possible to replicate the study. The use of a sediment grab for sample collection in the field and methodology (unclear) of sediment core collection and bulk density calculation is a weakness of the study, these should be clarified and the potential implications of using these should be appropriately discussed. The Methods should have a separate and condensed data analyses section, as it stands analyses are included within each section and that leads to continued repetition. There is a confusion in this manuscript as to what belongs in which section, with parts of the Discussion placed in the Methods and Results, and Methods in Results. The lack of environmental information from each location is a weakness in this study, as many studies have highlighted the effect environmental variables can have on OC storage. There is a recent paper by Gullstrom et al 2017 which presents insights on blue carbon from this region, including a sampling location (of nine in total) in 2012 at the same location sampled in the study for this manuscript. The authors cite this paper briefly, but it is critical that this manuscript clearly state how it is novel and how it differs from Gullstrom et al 2017. Furthermore, the variation among species in regards to OC has also been studied before, and therefore it is important to highlight what is novel in this manuscript, at the moment this is not clear from the text.

Specific comments:

Abstract

L2: what does "highly diverse mean"? it is not necessary and can be deleted

L3: delete "amount of"

Include how the sediment OC was quantified in the abstract – how deep were the cores and to what depth is the calculation being standardized

If word limit allows, consider including some basic biomass, density and N data in the abstract itself.

Introduction

P1L5-8: Seagrasses are now accepted as important carbon sinks; this idea needs to be reworked and the literature updated. I would suggest looking at recent work by Duarte, Macreadie, Marbá to start with.

P2L1-4: Previous research studying the link between seagrass species and plant characteristics needs to be described in greater detail to identify what the gap is that this manuscript would be filling. As it stands from the Introduction it would seem that seagrass sediment OC is known to vary with seagrass functional traits and environmental conditions and that blue carbon has already been assessed at the study location. From the introduction it seems that there is no novelty in this study other than studying this link at a new locations (Tanzania), which is not true based on Gullstrom et al 2017, and is a different aim than the one described. I would suggest greater focus on the functional trait approach.

P2L7-10: This needs clarification, yes there is variation in OC stocks given the different factors described in the previous paragraph and site-specific quantification of OC is needed, but there is no clear link between that fact and the "formable obstacle for reliably valuing the ecosystem service of OC". There is an idea missing here to link these two or greater clarification

P2L11: "fast-slow" is not common terminology used for the different life strategies of seagrasses, see Kendrik et al 2012 BioScience and Orth et al 2006 BioScience who use "ephemeral" and "persistent" or O'Brien et al 2017 MPB which uses "persistent", "opportunistic" and "colonizing". Orth et al 2006 is cited at the end of this section, but the terminology used by them is not included.

Aim paragraph: The aim needs to be modified, it does not appear to be to "identify where high sediment OC stocks occur" but whether three specific functional traits can be used as proxies for sediment OC content. There is no need to mention the five seagrass communities, focus on the question and the hypothesis. Rewrite.

Methods

A clear description of the characteristics that influence OC content in seagrass sediments at the three seagrass meadows sampled is needed, as they were mentioned in the Introduction and that information is lacking form the methods section, i.e. water depth, water clarity, hydrodynamics, geomorphic setting, etc.

P3L2: Cite Figure 1 here.

P3L5: change "warm and moist" to "tropical"

P3L6: when are the monsoon seasons?

P4. The use of a Van Veen Sampler if used for BC quantification is a weakness of the study, as this is not a method that reliably samples the exact volume or depth of sediment, it is greatly affected by the type of sediment and can be affected by the speed at which it drops. Was it used only for sediment characterization or for blue carbon quantification? at the moment this is not clear from the text, and collecting the "the upper 5-10 cm of sediment" is not a reliable method for quantifying sediment OC. Can you reliably say that the same depth was sampled at each site or is it possible that at some sites the grab collected more superficial sediment while at others it potentially collected deeper sediment? this is critical as OC content tends to decrease at greater sediment depths and given a general vertical accretion of 2 mm per year, you may have sampled completely different time periods. This needs to be adequately discussed as it is a key limitation of the study.

P4L15: "Surface sediments (top 2-3 cm) were also collected" how? using what? was the same processing protocol used?

P4L1&20: When in October?

P4L20-21: Describe the zonation

P3L21-22: How were the 50 m transects conducted? perpendicular to the coast line? consider including the transects in the figure. Change wording to: "A snorkeling survey was conducted at each meadow, consisting of five 50 m transects throughout each

meadow. Based on this initial survey, six to seven distinct vegetation zones were identified for each meadow."

P1L23: a quadrat of what size and for which purpose? If this is linked to the following paragraph consider merging the two.

P4L24-25: delete ", with the average distance of between sites of 261±194 meters for M1, 170±93 meters for M2 and 165±98 meters for M3"

P4L33-34: "square root transformed to down weight the influence of abundant species, and relativized to the total abundance of each site"

P4Section2.2: Change title to: "Seagrass community composition"

P5Section2.3: change title to "Seagrass functional traits"

P5L7 & L14: What do you mean by "seagrass plants"? Do you mean a shoot? a shoot with rhizome and root attached? a ramet?

P5L8: to what sediment depth was the core collected?

P5L8: why was leaf area not measured? it greatly affects seagrass cover and canopy structure.

P5L9: washed free of sediment with what? diameter of mesh is needed if one was used.

P5L11: change "number m-2" to "shoots m-2"

P5L15: What do you mean by "second-ranked"? from oldest or youngest?

P5L16: How did you remove epiphytes? acid, scraping? & what did you use to homogenize the samples?

I would suggest putting all stats in a separate section titled "Data analyses". Why were ANOVAs used instead of mixed effects models which would have accounted for the non-independence of the samples? If the %N cannot be statistically assessed

because of unequal and small sample sizes then you should only report it or exclude it from the manuscript. P6L13: a lot more detail on sediment coring methods is needed. For example, lacking information includes: dimensions and material of corer, how did you measure compaction from coring, etc. Below you mention that core depth went from 19-78 cm, that information should be up here not further down.

P6: there is an issue with the methodology for bulk density determination, if 15 ml from a core that has already been cut are being collected (by some undefined methodology) then the bulk density cannot really be reliable. Why did you not collect a specific volume of sample from the field from the first place and do bulk density on that?

P6L17: Acidification with what? % which acid

P6L18: include the units of CC

P6L28-29: not clear on why you state you only have N=18 as it was stated that there were three biomass cores collected at each location, that gives you (19 sites * 3) n=57. To avoid repetition, you should have a unified data analyses section.

Results

P7L7: what do you mean by "water clarity was high"? Provide data for comparison, for light attenuation too.

PL10: ", suggesting energetic hydrodynamic conditions" either you have data on hydrodynamic conditions or you are speculating, this should be adequately addressed or deleted.

P7L11-12: add standard deviations when providing means. No measure of variation is given in the supplementary material either, given that there are multiple samples per type of meadow this needs to be included. Stats can be done as well from my understanding of the sampling design, why haven't they been done?

P7L18 "using a combination of nMDS and hierarchical clustering" this should be

deleted from the Results, it should only be in the Methods.

P7L19-20. Delete this sentence, not needed.

P7L18-21: Should now read: "Five distinct species assemblages were identified, here-after referred to as communities A, B, C, D, and E."

P7L24: review comment on the appropriate terminology for seagrass life strategies instead of "fast-growing". Rephrase bc not all those species are "fast-growing"

P8L5: Should read "T. ciliatum" as it has already been presented in the previous paragraph, you could even just use TC as the abbreviation has already been mentioned as well.

P8L2-12 & L13-22: These paragraphs need to be rewritten to clearly state the biomass of each community and each meadow, do not try to discuss why one meadow had more than the other or one community more than the other, just state what you found. There is an inherent complication from assessing community and biomass which needs to be addressed in the Discussion, as there was no interaction the wording of the text also needs to reflect that.

P8L26-29: Why is there no standard deviation included for the density? and why does it say "(based on the negative binominal model)" & "estimated means"? From the text, the density was directly measured from the biomass cores, so if that data is directly available there should be no need to estimate it from a model and no problem with including standard deviations.

P9L4-5: "The entire range of leaf nitrogen content of communities A and B fell below the global threshold (1.82%) indicating nutrient limitation in seagrasses (Duarte, 1990)." this not belong in a Results section, move to Discussion if appropriate.

P9L10-12: "indicating the potential for nitrogen limitation and low microbial carbon-use efficiency during litter decomposition, both of which can lead to higher sediment OC sequestration (Berg and McClaugherty, 2003; Hessen et al., 2004)." this not belong in

a Results section, move to Discussion if appropriate.

P8L31-P9L16: The inclusion of N% needs to be carefully revised. It should only report the %N as the stats are very weak, and in my opinion, %N is not needed and can be deleted from the manuscript. It is also not adequately discussed in the Discussion, just delete it.

P9L18-19: "The depth that cores penetrated into the sediment varied from 19 to 78 cm and was dictated by the limited sediment accumulation on top of carbonate rock." This is not part of Results, move to Methods.

P9L19-25: If there is no variation in the top 25cm and core depth varied, then it cannot be said that "all cores exhibited the typical trend of decreasing %OC with depth into the sediment". This needs to be clarified, I assume it only refers to cores deeper than 25cm? how many cores were deeper than 25cm? that information is not presented in the text.

P9L25-29: "This indicates that the bare areas may have been colonized by seagrass in the past, contributing to an increase in carbon storage within deeper layers of the sediment. Thus, it must be noted that in order to associate present seagrass communities with long term carbon storage in sediments, we assumed there were no historic differences in communities during past carbon deposition." This does not belong in a Results section, any assumptions of past seagrass presence should be clearly stated in the Methods and discussed in the Discussion. Delete from here.

P9L30: What are you referring to when using the term "OC storage"? clarify the term.

P10L1-3: "Model validation of normality (Shapiro Wilks test) were met for all OC models (Supplementary Table S2), and variogram plots of model residuals showed no clear patterns indicating that the assumption of independence was met (Supplementary Figure S4)." This does not belong here, it belongs in the Methods, there is a clear confusion in this manuscript as to what belongs in which section.

Discussion

First paragraph: Restructure to remind the reader first of the link between traits and OC storage, then go into what was found overall. The following text can be deleted from here: "We hypothesized that communities with either high shoot density, low tissue nitrogen content, or a high proportion of belowground biomass would store more OC within their sediments. From this, it would be expected that community B (dominated by Thalassodendron ciliatum), with combined traits of high AG and BG biomass and low tissue nutrient content, or community C with high shoot density in two of the three meadows sampled would store more sediment OC.". Again, use adequate terminology for seagrass life strategies.

P10L21: Change "must" to "may". In the introduction it was clearly mentioned that there are a number of environmental variables that can affect OC storage, so this is not a finding from this study, you should refer back to the published literature on the topic.

P11L2-4: move this last sentence to be the first (topic) sentence of that paragraph, edit accordingly.

P11L11-14: Gullstrom et al 2017 is a blue carbon study which included a sampling location (of nine in total) in 2012 at the same location sampled in the study for this manuscript. It is critical that this manuscript clearly state how it is novel and how it differs from the Gullstrom et al 2017 paper in, which has already presented OC storage insights from this region, including the site sampled. The variation among species in regards to OC has also been studied before, and therefore it is very important to highlight what is novel in this manuscript, at the moment this is not clear.

P11L14-15: "Because most seagrass species occur at all locations, the contrast in OC storage among sites is likely influenced by differences in the depositional environment and/or sediment." This is clear from a number of studies now, why was no information collected on how the environmental characteristics vary at each location? This a weakness of the study.

The limitations of the study need to be discussed, for example, how many years of carbon burial do the cores represent? potential effects of coring methodology? data which is lacking? & suggestions for future studies?

Conclusion

Why are figures and literature cited in the conclusion? This conclusion needs to be incorporated in the Discussion itself adequately.

Figures

In Figure 1 M1, M2 and M3 are differentiated by three different colours (yellow, green and red), I would suggest that these same three colours are used for the figures that refer to meadows 1,2,3 differentiating them from the figures that refer to the different communities (A,B,C,D,E,F). As it stands, from the colour scheme it seems that you are referring to communities when in fact you are differentiating the communities according to the meadow. I would also suggest not using green and red together, as that is not easily distinguishable for colour blind (Daltonism) readers, may I suggest the three primary colours, changing the light green for a light blue? whichever colours are chosen, please be consistent among the figures.

Supplementary

I would not include any of this as supplementary material, except Table S1 in the main text. If it is important information then include it in the main text, if not then delete it. Needed stats should be incorporated into the text but a lot of that information and figures are simply not needed.

Technical corrections:

P3L4-6 & P11L1: There should be no italics for "spp."

P3L7: Change "objective" for "aim"

P3L28: Delete "," after several

P3L29: Typo should be "west of"

P4L2-3: Change "50-m transects" to "50 m transects"

P4L23: should be six to seven, not 6 to 7

P4L29: ", and" should not be in italics

P5L1: "nonmetric" should be "non-metric"

P6L17: "3-cm" should be "3 cm"

P9L32: "1 meter" should be "1 m"

---

## Referee Comment (RC2) · Anonymous Referee #2 · 26 Jan 2018

**Overall comments**

The manuscript by Belshe et al. attempts to provide insights into blue carbon storage capacity in seagrass areas off Zanzibar Island. While I view the study a welcome addition to the increasing global and regional focus on blue carbon sequestration, the style of argument/discussion places the authors' findings in a negative light rather than a substantial progress in this field. I have elaborated on this matter below, and other suggestions that will improve the manuscript. I look forward to reading a revised version of the manuscript in the near future.

Specific comments

1) One of the shortfalls in Fourqurean et al. 2012's paper is the limited number of African meadows considered in that paper's meta-analysis. This current study complements those already done off the African continent and would therefore allow more robust regional estimates in OC sequestration capacity. The authors, however, reported and emphasized low OC stocks in their study sites. This is not novel, in my opinion, since the authors' 33.9 Mg C ha-1 estimates: 1) still fall in the global range of 9-628 Mg Corg ha-1 in Fourqurean et al. 2012; 2) is just slightly higher than Fourqurean et al. 2012's estimate for the Indo-pacific region of 23.6 Mg Corg ha-1; and 3) not that much different to those estimates done in SE Asia, which is in the same bioregion as this study (see below on this, and please also refer to OC stock estimates in Miyajima et al. 2015; Gilis et al. 2016, Quak et al. 2016; Rozaimi et al. 2017).

2) There is a fixation by this study as well as others already published on trying to predict OC storage capacity by biological and or physical drivers. It has already been suggested in Lavery et al. 2013 that variability can be expected and therefore I don't find it surprising Gullstrom et al. 2017 had different results compared to this study. Furthermore, many studies, e.g. Serrano et al. 2016a, already connected sediment grain size as negatively correlated with OC stocks. I don't see the logic, therefore, to persist looking into such "geophysical constraints" whereupon low OC stocks are to be expected. As it stands, this is the angle that the authors communicated, and therefore I view this study's findings not very interesting. BG is rarely seen as a journal of negative results and I recommend the authors portray their findings in a different light. What I do find interesting is that high seagrass biomass/density does not necessarily translate to high sediment OC stocks, especially in reference to the study's findings in Community B. Indeed, this is in stark comparison to e.g. Macreadie et al. 2012&2015's, and Serrano et al. 2014&2016's Posidonia studies, where such correlation is expected. I believe this angle can pique the interests of BG readers more than how it is now.

3) As it stands, using %N as a predictor variable is a particularly weak approach.

Seagrasses are naturally N limited and therefore I don't see its justification in this study. I note the authors attempted to relate N content/decomposition to CNP stoichiometry (Pg 2 L 29-31) but I need further convincing before agreeing this as a viable approach.

4) I am not comfortable with the authors' way in presenting OC cycling as the alternative explanation for their findings of low OC stocks. The context linking sequestration capacity and OC cycling is too broad in the absence of sufficient evidence, which in turn made it a rather unconvincing discussion. I suggest the authors discuss the findings along the lines that the studied meadows have low capacity to sequester autochthonous inputs. Such approach would still be in the bigger auspices of OC cycling and will not stray too far from the body of evidence already presented.

5) There is an underlying initial assumption by the authors that seagrass tissues are buried in the sediment and therefore sequestration occurred. It is unfortunate that OC provenances did not fall into the scope of this study. I do not insist this be done, but nonetheless, it is reasonable to infer from the results in Miyajima et al. 2015; Gilis et al. 2016, Quak et al. 2016 and Rozaimi et al. 2017 on seagrass endmember contributions to OC sediment sequestration. The seagrass meadows in these four studies (with the exception of particular sub-tropical and temperate meadows in Miyajima et al. 2015) and those in this study, are in the same bioregion (after Fourqurean et al. 2012). The studies I quote reported low seagrass contributions and also low OC stocks to the sediment. I also refer the authors to Bouillon et al. 2004 for a data set on OC provenances that were obtained closer to their sites. These papers may assist the authors in arguing their case succinctly.

6) I am particularly uncomfortable with the supposition laid in Pg 9 L25-28 on the historical colonization of seagrasses in the study area vis-à-vis carbon deposition and seagrass community structure. It is not supported by historical data and/or organic matter provenances (see above). The authors should have considered that the period of carbon accumulation (re Serrano et al. 2016b) is an important aspect of blue carbon accounting. A stronger case is needed before readers would agree to the assumption

posed by the authors.

7) I recommend adding a Figure or Table summarizing the OC density (g OC cm-3) and/or sediment dry bulk density data to complement Figs 6 and 7.

Methods and design

8) General comments: a) specify water column depths of the sampled sites; b) specify if epiphytes were removed before weighing biomass samples for above-ground plant parts; c) clarify sediment acidification protocols; d) specify the use of CN ratio calculations in methods.

9) Coring methodology: a) I can accept that core compaction during sampling can be assumed negligible for short cores but longer cores require core length corrections. Please refer to Howard et al. 2014; b) I find it perplexing that the authors included extremely short cores in analysis when in fact it was possible to get longer cores within the same community site (i.e. A, B, C and F). The only logic I fathom is the insistence on a replication/ecological approach, which is not particularly essential in these types of biogeochemical/biogeophysical studies. Such inclusions of short and long cores as "replications" invites greater variability and more questions are thus raised on the robustness of the findings.

10) General writing clarity Pg 3 L16-18: "Yet, the ... Gullstrom et al. 2017) - Does not fit in Methods. Either move or remove to the appropriate section. Pg 9 L10-12: "...indicating the... Hessen et al. 2004)." Does not fit in Results. Either move or remove to the appropriate section. Pg 10 L5-8: Break this into two sentences Pg 10 L30-31: "Some of ... biomass" - Unclear sentence structure. Please edit. Pg 11 L15: ...and/or sediment <what?> - Sentence appears hanging. Pg 12 L29-30: A general rule I follow is to avoid references in conclusions, which would otherwise infer weak arguments for a study. Figure captions: Please consider truncating the captions to relaying the most relevant information only.

[Figure]

11) Minor technical edits Pg 1 L13: is sediment -> in sediment Pg 2 L25: determinates -> determinants Pg 4 L7: Sedimentary samples -> Sediment samples Pg 6 L20: g OC per dry weight <sediment?> Pg 7 L13: granumetrical -> granulometric Pg 13 to 20: Please relook at the reference list thoroughly: a) spacing between words, italicizations of species and genus names that are lacking should be edited; b) edit Serrano et al 2016 to Serrano et al. 2016a and 2016b, and the latter was already accepted as BG and should not be a BGD citation; c) you should cite the more recent Costanza et al. 2014 paper, rather than the 1997 paper; d) consider updating/revisiting the reference list as suggested in this review:

Bouillon et al. 2004. et al. Biogeosciences 1, 71–78. Costanza et al. 2014. Glob. Environ. Chang. 26, 152–158. Fourqurean et al. 2012. Nat. Geosci. 5, 505–509. Gillis et al. J. Sea Res. 120, 35–40. Gullström et al. 2017. Ecosystems 1–16. https://doi.org/10.1007/s10021-017-0170-8 Howard et al. 2014. Coastal Blue Carbon: Methods for assessing carbon stocks and emissions factors in mangroves, tidal salt marshes, and seagrass meadows. Macreadie et al. 2012. Glob. Chang. Biol. 18, 891–901. Macreadie et al. 2015. Proc. R. Soc. B Biol. Sci. 282. Miyajima et al. 2015. Global Biogeochem. Cycles 29, 397-415. Quak et al. 2016. Estuar. Coast. Shelf Sci. 182, 136–145. Rozaimi et al. 2017. Mar. Pollut. Bull. 119. 253-260. Serrano et al. 2014. Global Biogeochem. Cycles 28, 950–961. Serrano et al. 2016a. Biogeosciences 13, 4915–4926. Serrano et al. 2016b. Biogeosciences 13, 4581–4594.

---

## Author Comment (AC1) · 23 Feb 2018

The comment was uploaded in the form of a supplement:
https://www.biogeosciences-discuss.net/bg-2017-474/bg-2017-474-AC1-supplement.zip

[Figure]

[Figure]

**Fig. 1.** Figure 2: Average percentage of sediments within each sediment grains size class (top left) for the four biogeographic zones (tidal channel, fore reef, reef flat, and seagrass meadow).

---

## Author Comment (AC2) · 23 Feb 2018

The comment was uploaded in the form of a supplement:
https://www.biogeosciences-discuss.net/bg-2017-474/bg-2017-474-AC2-supplement.zip

---

## Author Response (AR1)

Dear Dr. Peter von Bodegom

We are very grateful for the chance to resubmit our manuscript. Although we must go back through the review process, we believe this updated version of our manuscript has been greatly improved by the previous reviews.

As to the changes we made from our original submission (that went to review), we incorporated all of changes we proposed in response to the comments of Anonymous Referee #2. These specific changes were detailed in our replies to Referee 2, please let us know if you need these again. We have also incorporated most of the changes in response to comments of Anonymous Referee #1 but have taken your advice and did not include all suggested changes to the introduction, specifically in regards the addition of the following:

"A suite of studies have shown that larger OC stocks can be found in sediments under seagrasses with conservative 'slow' plant traits, such as high above and belowground biomass (Armitage and Fourqurean, 2016; Dahl et al., 2016; Gullström et al., 2017), and low-quality tissues (Kaal et al., 2016; Serrano et al., 2016b; 2012; Trevathan-Tackett et al., 2017). OC stocks have also been positively correlated with shoot density, both directly (Dahl et al., 2016) and indirectly (Serrano et al., 2014). However, at global and regional scales, when comparing OC storage across disparate sites, the explanatory power of plant traits can be overshadowed by abiotic factors, such as differences in sediment properties and water flow regimes (Campbell et al., 2014; Dahl et al., 2016; Lavery et al., 2013; Serrano et al., 2016a). Although the largest OC stores are found within meadows of seagrasses with 'slow' plant traits (Fourqurean et al., 2012b; Serrano et al., 2012; 2016a), further evidence is needed to confirm the universality of plant traits as proxies for sediment OC content within a given site."

We believe this placed too much of the discussion into the introduction and made our introduction very long. However, if this was not what you were referring to please let us know and we can add this back to the last paragraph of the intro where we indicate the aims of our study. Likewise, if you need the specific changes to the manuscript made in response to comments by Referee 1, they are detailed in our previous response but can provide them again if needed.

Thank you again for this opportunity and we look forward to any further improvements that can be gained by the additional round of reviews.

Kind regards, E. Fay Belshe

---

## Referee Report (RR1)

In regards to the resubmission by Belshe et al., the revisions had significantly improved the manuscript. My comments were fairly addressed and the authors presented an acceptable balance between agreeing to, or arguing against my list of queries. I have no further comments for revisions and recommend the manuscript's acceptance for publication.